# Quality of cowpea seeds: A food security strategy in the tropical environment

**Leticia de Aguila Moreno**[◉], **Gustavo Roberto Fonseca de Oliveira**[◉], **Thiago Barbosa Batista**[◉], **João William Bossolani**[◉][‡], **Karina Renostro Ducatti**[‡], **Cristiane Carvalho Guimarães**[‡], **Edvaldo Aparecido Amaral da Silva**[◉]*

Crop Production Department, College of Agricultural Science, State University of São Paulo (UNESP), Botucatu, São Paulo, Brazil

◉ These authors contributed equally to this work.
‡ JWB, KRD and CCG also contributed equally to this work.
* amaral.silva@unesp.br

**Data Availability Statement:** All relevant data are within the paper and its Supporting Information files.

## Abstract

What is the relation between seed quality and food security? Here we built a summary diagram that links the development stages of the seeds with their potential of providing grain yield. This idea was tested using cowpea as a model crop, grown in a tropical environment. Initially, seed quality attributes such as water content, dry weight, germination, vigor, and longevity were characterized during seed development. With this, we were able to elucidate at which point the late maturation phase and the acquisition of seed with superior physiological quality starts. From these data, the proposed summary diagram highlighted the seed quality as a technological basis for generating a more productive plant community. It also showed that only seeds with a high-quality profile have a better chance to establishment in an increasingly challenging agricultural environment. Overall, we bring the concept that cowpea seeds with superior quality besides being the essential input for tropical agriculture is also a strategy that can contribute food security.

## Introduction

The cowpea plant (*Vigna unguiculata* L., Walp) is a legume resilient to the adversities of the agricultural environment, being commonly cultivated in regions with high temperatures and water restriction [1]. The grains have several food uses, providing high-energy compounds that benefit human health and combat malnutrition [2, 3]. An adequate stock of cowpea is important for the subsistence of societies at nutritional risk, especially on the African continent, where these grains are consumed on a larger scale [4]. The exploration of factors which can improve the yield of cowpea plants is part of a broad food safety practice. Thus, investigating when cowpea seeds acquire maximum quality is an opportunity to collaborate with this purpose.

Seeds are one of the greatest assets of agricultural activity in the world. Their physiological quality defines both the establishment of the crop and its productivity [5, 6]. High-quality seeds are more capable of generating uniform seedlings that will give rise to individuals with

**Funding:** This study was supported by: Coordenacão de aperfeiçoamento de Pessoal de Nível Superior (CAPES) (Financial Code 001) to L. A. Moreno, Conselho Nacional de Desenvolvimento Científico e Tecnológico (grant number 311526/2021) to E.A. Amaral da Silva. The funders had no role in study design, data collection and analysis, decision to publish, or preparation of the manuscript.

**Competing interests:** The authors have declared that no competing interests exist.

ample productive potential and efficient use of the resources of the environment [7]. On the other hand, seeds with reduced quality will compromise the emergence of seedlings, and result in an uneven establishment in the field, which in turn will result in low yield [8, 9]. Faced with the context of climatic stress where cowpea is most produced, we pose the following question: when do cowpea seeds acquire maximum physiological quality? Thus, the time when cowpea seeds are harvested can increase the chances of high seedling establishment under wide environmental stresses, which is the case of where cowpea is normally cultivated.

The connection of the seed to the food production process starts from the expression of quality attributes acquired during its development [10]. These attributes establish a bridge to a new plant cycle in the field. In particular, germination, desiccation tolerance, vigor and longevity are acquired during the maturation and late maturation phases [11]. Interestingly, the moment of acquisition of these attributes varies among plant species [12], which makes its understanding specific for each crop of agricultural interest. In other legumes such as soybeans [13], *Medicago truncatula* [14] and peanuts [15], efforts have been made to understand the acquisition of physiological quality. This made it possible to determine when the seeds complete their development and acquire their maximum quality. In cowpea seeds, there is still a knowledge gap in the subject, which may limit the technological advances expected for its cultivation, especially in regions of the world commonly under severe climatic stresses.

The use of cowpea seeds with superior quality can be strategic for optimizing grain yields in regions that naturally have scarce resources for crop production. The purpose of this work was to characterize the acquisition of physiological quality of cowpea seeds and to propose a connection of this process with plant yield. Furthermore, we are creating a general idea that seed quality is a strategy for food security in tropical environments.

## Material and methods

### Plant material and local

The commercial cowpea seeds (cultivar BRS Guariba) were produced under field conditions from April to August during the crop season of the year 2015. The research was performed in an experimental area of the College of Agricultural Sciences, State University of São Paulo (UNESP), Botucatu, São Paulo, Brazil. All necessary approvals were obtained for the study, which met all relevant regulations.

### Trial design

The development of cowpea seeds was monitored under tropical growing conditions. We characterized the maturation and late maturation phases of seeds based on days after anthesis (DAA). For this, attributes such as water content, dry weight, vigor, and longevity of seeds were measured over the time of seed development. This data allowed us to define the moment to harvest cowpea seeds with the highest quality. It also allowed the construction of a summary diagram with application to agricultural species propagated by orthodox seeds. In this diagram, we connected the phases of acquisition of physiological quality with the establishment of plants in the field and grain yield. The knowledge acquired in this work, added to that existing in the literature, brings the idea that enhancing the quality of cowpea seeds increases its relevance in food security strategies.

### Characterization of the seed production area

The soil was characterized as an Oxisol [16]. The area was subjected to soil preparation (plowing and harrowing) and sowing fertilization (20 kg ha$^{-1}$ of simple superphosphate [46% $P_2O_5$]

and 20 kg ha$^{-1}$ potassium chloride [K$_2$O]) according to the chemical attributes of the soil collected from the 0.0 to 20.0 cm layer (pH CaCl$_2$: 5.0; Organic matter: 23.0 g dm$^{-3}$; P resin: 37.0 mg dm$^{-3}$; S: 23 mg dm$^{-3}$; Al$^{3+}$: 0.0 mmolc dm$^{-3}$; H+Al$^{3+}$: 43.0 mmolc dm$^{-3}$; K: 2.7 mmolc dm$^{-3}$; Ca$^{2+}$: 28.0 mmolc dm$^{-3}$; Mg$^{2+}$: 12 mmolc dm$^{-3}$; sum of bases: 43.0 mmolc dm$^{-3}$; CTC: 86 mmolc dm$^{-3}$; base saturation: 50%).

## Plant cultivation

The seeds were inoculated with *Bradyrhizobium* spp. (at a rate of 100 mL of inoculant per 100 kg of seed), and sowing was performed on April 18th, 2015, at a density of ~18 viable seeds m$^{-1}$ at rows spaced 0.70 m apart. In addition, covering fertilization was done at 15 and 30 days after seedling emergence, using 20 kg ha$^{-1}$ of N, in the form of ammonium sulfate. Seedling emergence occurred on April 23rd 2015 and the appearance of the first floral buds occurred on July 13th 2015. Seed harvesting was then started after that date (17 DAA: July 29th 2015) and ended on August 21st, 2015 (40 DAA). The experiment was conducted under dry conditions, without water supplementation throughout the crop cycle. The cycle lasted 120 days from sowing to final harvest. The average maximum and minimum temperatures during this period were 24˚C and 15˚C, respectively. Additionally, the average relative humidity (RH) was 74.3%, and the total rainfall was 286 mm (S1 Fig).

## Characterization of seed maturity stages

The characterization of seed maturity stages was based on the work done for soybean [17] with modifications for cowpea seeds. Its flowers remain open only for a few hours in the morning, and when they close, they already initiate the development of the seed. Therefore, the flowering tag was made early in the morning (S2 Fig). Flowers were tagged at the day of opening, in order to track the development of the fruit and determine the number of days after anthesis (DAA). Fruits were collected manually from 17 DAA to 40 DAA. The seeds were extracted manually from the fruits and used immediately for analyses of dry weight, fresh weight, and water content, totaling five reproductive stages. According to germination pre-tests performed on different days after anthesis, it was found that the seeds were able to germinate only 28 DAA (S1 Table). Therefore, for the tests of germination, vigor, tolerance to desiccation and longevity only the seeds in the last five reproductive stages were used, that is, from 28 DAA to 40 DAA (at 3-day intervals).

## Seed quality assessment

**Water content.** The water content of fresh (freshly harvested) seeds was determined by drying four repetitions of 10 seeds in an oven at 105˚ C constant for 24 hours [18]. The water content obtained was expressed as a degree of humidity on a wet basis.

**Dry weight.** For the determination of dry weight (DW), four repetitions of 10 fresh seeds were dried in an oven at 60˚ C for 72 hours. The DW results were expressed in milligrams per seed.

**Germination capacity.** The germination capacity of the seeds (fresh and non-dried seeds) was performed at 28, 31, 34, 37 and 40 DAA. Three replicates of 25 seeds from each maturation stage were placed on germination paper moistened with water equivalent to 2.5 times the weight of the paper, arranged in an upright position at a temperature of 20˚ C in a germinator. The radicle protrusion was evaluated 120 hours after test installation.

**Desiccation tolerance.** To determine the acquisition of desiccation tolerance, immediately after harvest the seeds were distributed in single layers on the surface of a wire mesh screen (10.5 x 10.5 x 1.8 cm) suspended over silica gel inside a plastic box (11.0 x 11.0 x 3.5 cm)

at 20˚C, to dry gradually until they reached a moisture content of 10% (wet basis) [19]. After this, the germination test was performed with three repetitions of 25 seeds each. Seeds with a radicle protrusion of at least 2 mm of length were considered desiccation tolerant (evaluated 120 hours after test installation).

**Germanation rate.** During the germination test (dried seeds), we also evaluated seed vigor by determining the germination rate (1/t50). The measurements of time required for 50% of the seeds to germinate (t50) were taken every 6 hours until 8 days. Seeds with a radicle protrusion of at least 2 mm of length were considered germinated. The results were analyzed using the curve adjustment of the Germinator software [20].

**Seedling formation.** In parallel, we also measured seed vigor by calculating the seedlings formation at five days after the beginning of the germination test. We considered normal seedlings those with well-developed essential structures, (aerial part, hypocotyl and radicle) [18].

**Seed longevity.** For the determination of longevity, 150 seeds for each moment analyzed (28, 31, 34, 37 and 40 DAA) were dried as previously described, and distributed in single layers on the surface of a wire mesh screen (10.5 x 10.5 x 1.8 cm) suspended over a NaCl saline solution (75% RH) inside a plastic accelerated ageing box (11.0 x 11.0 x 3.5 cm) at 35˚ C. The seeds with 28 DAA were removed from the plastic boxes at 3-day intervals and the germination test was performed. For the remaining DAA (31, 34, 37, and 40), the seeds were removed at 10-day intervals to perform the germination test describe above. For this analysis, a radicle up to 2 mm of length was considered as the criterion for germination. The storage data of seeds from 28, 31, 34, 37 and 40 DAA were fitted with sigmoid $f = y0 + a \, (1 + e^{(-(x-x_0)/b)})$, to determine the moment when the initial germination was reduced by half (p50), represented by its intersection with fitted sigmoid.

**Duration of the maturation and late maturation stages.** The percentage duration of the maturation and late maturation stages of cowpea seeds was calculated based on the proportion in days of each stage to the total development time of the seeds after DAA. Seeds began to be harvested at 28 DAA, which was considered the beginning of the maturation stage. Harvesting continued until 40 DAA, which was considered the end of the late maturation phase and the end of seed development.

## Summary diagram

The summary diagram was designed from the observed results for water content, dry weight, desiccation tolerance, vigor (1/t50), and longevity (p50) during their maturation and late maturation (28, 31, 34, 37 and 40 DAA). The measuring of attributes of physiological quality made it possible to determine the moment of acquisition of maximum quality for future crop formation. Thus, we connected the seed developing stages with the grain yield of cowpea. The conclusion is that enhancing the quality of cowpea seeds creates opportunities for new food security strategies.

## Statistical analysis

The experiment was carried out on a completely randomized design with five maturation stages (28, 31, 34, 37 and 40 DAA) as a source of variation with three replications for each stage (n = 15) with 20 seeds per replication. The data were transformed (Box Cox transformation) as necessary to meet the assumptions of the analysis of variance (ANOVA). The data obtained at each reproductive stage were subjected to ANOVA and the Tukey test at a significance level of 5%. The "ExpDes.pt" package of the R software was used to perform this analysis [21]. The seed quality parameters such as water content, dry weight, germination capacity, desiccation tolerance, germination rate and seedling formation were subjected to principal

components analysis. For the respective analyses, One-way PERMANOVA [22] was used to group treatments by similarity (Bray-Curtis similarity index) to identify significance between the groups obtained according to the maturation stages at a significance level of 1% (software Canoco 5).

## Results

Seed moisture at the beginning of development was high, where those with 28 DAA had a water content above 60% (Fig 1A). There was a decrease in moisture content during seed maturation (Table 1). The dry weight of the seeds started to increase from 7.3 mg/seed at 28 DAA to 12.01 mg/seed at 34 DAA (Table 1). The seeds capable of germinating at 28 DAA had a water content of 64% and, throughout development, this value decreased to 50% (34 DAA), until reaching 9% at the end of the late maturation phase (40 DAA) (Fig 1A).

Germination capacity had not been yet acquired at 25 DAA (S1 Table). This capacity reached maximum potential at 37 DAA (Fig 1B). Therefore, our results confirmed that mass maturity can't always be considered the moment when the seeds have superior physiological quality, since seed quality was still being acquired after the maximum accumulation of dry weight (34 DAA).

Seeds with less than 28 DAA were not able to germinate after drying (Fig 1B). This information indicates that they had not yet acquired desiccation tolerance (Table 1). At 28 DAA, the seeds began to acquire desiccation tolerance (11% of germination after drying), reaching a maximum value at 31 DAA, when they presented the highest germination (Fig 1B). In parallel, the results of vigor expressed by the germination rate (1/t50) showed that the seeds at 37 DAA. Concomitantly, the seedling formation increased from 28 DAA up to 40 DAA (Fig 1C). Although germination was already considered high at 31 DAA, the production of normal seedlings showed reduced values (7%). The maximum development of normal seedlings (seedling formation) occurred only at the 40th DAA, with 81% (Fig 1C).

In our study, while water content decreased, and desiccation tolerance was progressively acquired (Fig 1A and 1B). The ability of the seeds to remain viable during storage also increased (Fig 2A). During storage, seeds with 28 DAA did not tolerate more than 20 days of storage (Fig 2A). With 31 DAA, the cowpea seeds had already acquired greater storage capacity (44 days). As the seeds developed, the storage capacity increased, as can be seen in seeds with 34 DAA (98 days), 37 DAA (147 days) and 40 DAA (156 days) (Fig 2B). Thus, seed longevity was gradually acquired from 28 DAA to 40 DAA. In the beginning of the seed maturation phase, cowpea seeds (28 DAA) can not be stored for a long period. It is important to mention that the level of vigor increased concomitantly with the acquisition of longevity (Fig 1C).

From the principal component analysis (PCA), the distinction between the seed maturation stages was verified at 1% significance. The data set observed for seeds harvested 37 DAA and 40 DAA were grouped in the same quadrant, presenting the same trend (higher modulus of the vector) with the highest values obtained for p50 (longevity) and SE (vigor). Overall, cowpea seeds at late maturation stages showed the highest physiological quality (Fig 3). In addition, the late maturation stage was found to represent around 10% of the total developmental period of cowpea seeds (Fig 4). From the presented results, the summary diagram was built. It allowed connecting the physiological behavior of seeds with their likely performance in plant establishment in the field and grain yield. Overall, the constructed idea highlights that the use of cowpea seeds harvested at 37 DAA and 40 DAA (high quality seeds) provide a higher chance of generating a more productive plant community (Fig 5).

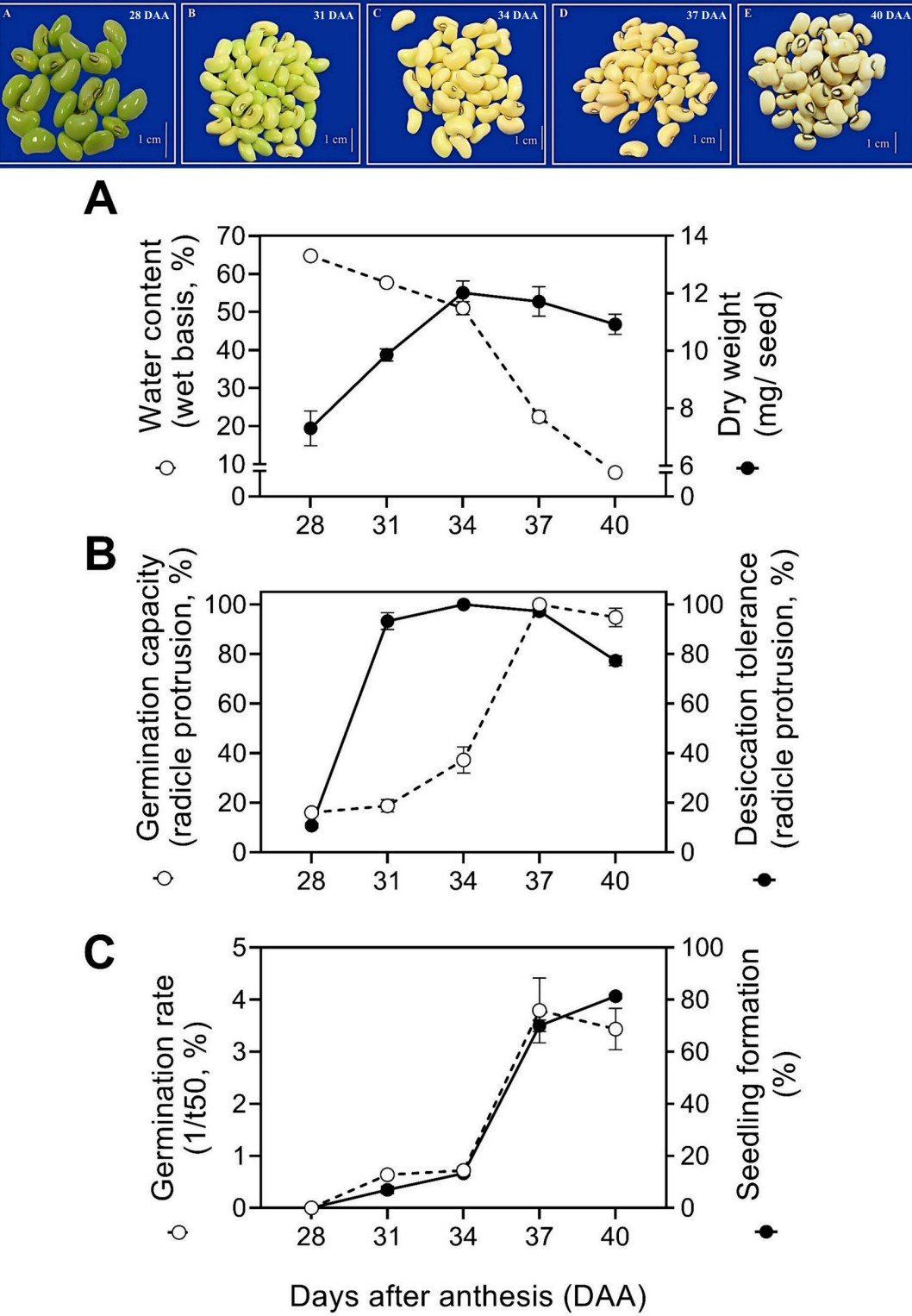

**Fig 1. Physical and physiological changes of cowpea (*Vigna unguiculata* L., Walp) seeds at 28, 31, 37 and 40 days after anthesis (DAA).** A: Water content and dry weight. B: germination capacity (fresh seeds) and desiccation tolerance (dried seeds). C: vigor expressed by germination rate (1/t50) and seedling formation (aerial part, hypocotyl and radicle well developed and healthy).

**Table 1. Statistical analysis.** Information of the data evaluated in cowpea seeds (*Vigna unguiculata* L., Walp) during maturation stages (28 to 40 DAA)*.

| DAF | WC [1] | DW | G | DT | 1/t50 | SE |
|---|---|---|---|---|---|---|
| 28 | 64.71 a | 7.30 c | 16.0 c | 10.0 c | 0.00 b | 0.0 e |
| 31 | 57.66 b | 9.86 b | 18.0 c | 93.0 b | 0.63 b | 7.0 d |
| 34 | 50.96 c | 12.01 a | 37.0 b | 100.0 a | 0.72 b | 13.0 c |
| 37 | 22.48 d | 11.71 a | 100.0 a | 97.0 b | 3.79 a | 70.0 b |
| 40 | 9.44 e | 10.92 a | 94.0 a | 77.0 b | 3.43 a | 81.0 a |
| LSD | 5.62 | 1.91 | 3.14 | 3.84 | 8.02 | 5.67 |
| C.V.% | 6.27 | 8.46 | 11.94 | 4.86 | 38.59 | 7.58 |
| p value | 0.001 | 0.001 | 0.001 | 0.001 | 0.001 | 0.001 |

*WC: water content (%); DW: dry weight (mg/seed); G: germination capacity (radicle protrusion, %); DT: desiccation tolerance (radicle protrusion, %); 1/t50: germination rate (%); SE: seedling formation (5 days, %); p50: longevity seeds during maturation expressed in p50 (period in days when the viability of the seeds is reduced by 50% viability). * Averages followed by the same lower case letter in the column do not differ by Tukey test at 5% probability.

## Discussion

Here we show the process of developing cowpea seeds to achieve maximum physiological quality in a tropical environment. We propose a summary diagram that highlights the acquisition of seeds with superior quality as a technological basis for the formation of plants with high yield. Overall, we bring the idea that harvesting high-performance seeds is a strategic part of food security programs.

We began our study by showing the physical changes during seed development. Cowpea plants generate seeds with reported orthodox behavior [23], as they start development (28 DAA) with high water content (70%) and gradually dehydrate with the advancing of the maturation phase (10%). The high moisture content in the early stages (28 DAA and 31 DAA) is known to support the division and expansion activities of embryonic cells [24]. Additionally, it allows the translocation of assimilates to the reserve tissues of the developing embryo [25]. As seed moisture content and metabolism are reduced (Fig 1A), the accumulation of reserve compounds is established and determines mass maturity (34 DAA). Interestingly, in cultivated

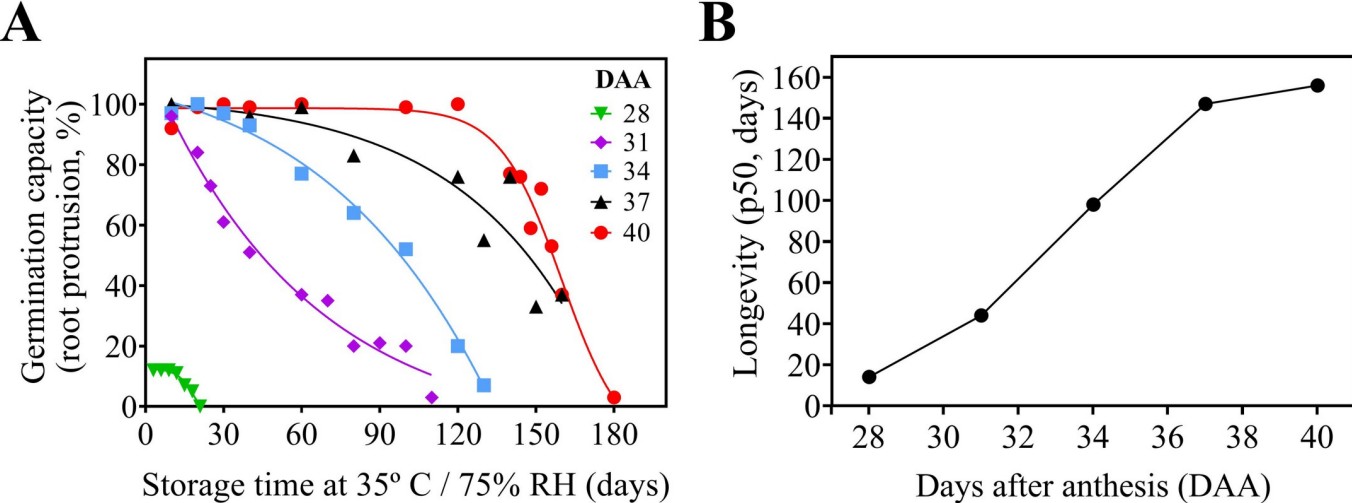

**Fig 2. Lifespan of the cowpea (*Vigna unguiculata* L., Walp) seeds harvested at 28, 31, 34, 37 and 40 DAA.** A: Germination capacity after different periods of storage. B: acquisition of seed longevity expressed by p50 (period in days when the viability of the seeds is reduced by 50%).

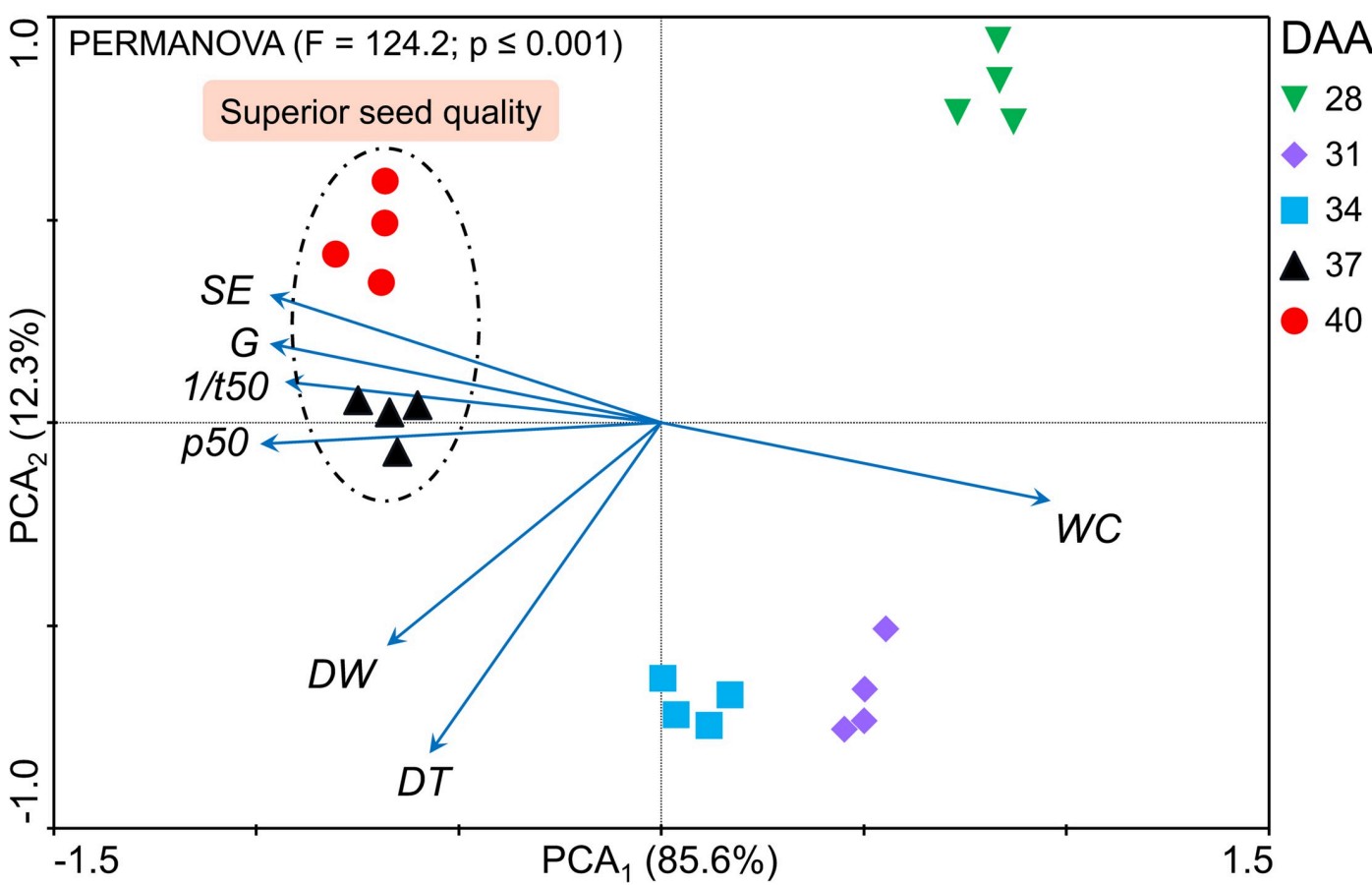

**Fig 3. Principal component analysis (PCA).** The dotted circle indicates the variables that most correlated at 37 DAA and 40 DAA (superior seed quality: late maturation phase). WC: Water content; DW: Dry weight; G: Germination capacity; DT: Desiccation tolerance; 1/t50: germination rate; SE: seedling formation; p50: Longevity.

species it has been shown that maximum seed filling (mass maturity) is a fundamental part of the development of leguminous seeds. However, it is not the end of their maturation process [12]. Here, we reinforce that those physical changes in cowpea seeds occur together with other events that support the acquisition of physiological quality attributes.

We found that the germination capacity of cowpea seeds was the first attribute to be acquired (Fig 1B). In species such as soybean [26] and *Medicago truncatula* [14], germination *sensu stricto* (radicle protrusion) still occurs under high water content. However, corroborating the behavior of these species, we found that this initial germination potential is incipient when the seeds are exposed to drying (Fig 1B). Also, as reserve compounds become part of the cotyledons, desiccation tolerance is installed, and the seeds resist intense water loss staying alive (Fig 1A and 1B). Studies have sought to elucidate the mechanisms that support this behavior in leguminous seeds associated with the accumulation of reserves such as LEA proteins [19], HSPs [27] and oligosaccharides [26] during the seed filling phase. The essential role of these compounds in the acquisition of desiccation tolerance ensures the survival of the seeds in the dry state [23]. In cowpea seeds, they mark the maturation phase and provide support for the progress for the complete acquisition of the seed physiological quality.

Cowpea seeds, once they acquired mass maturity, germination capacity and desiccation tolerance, continue to gather physiological competences associated with vigor (Fig 1C). Indeed,

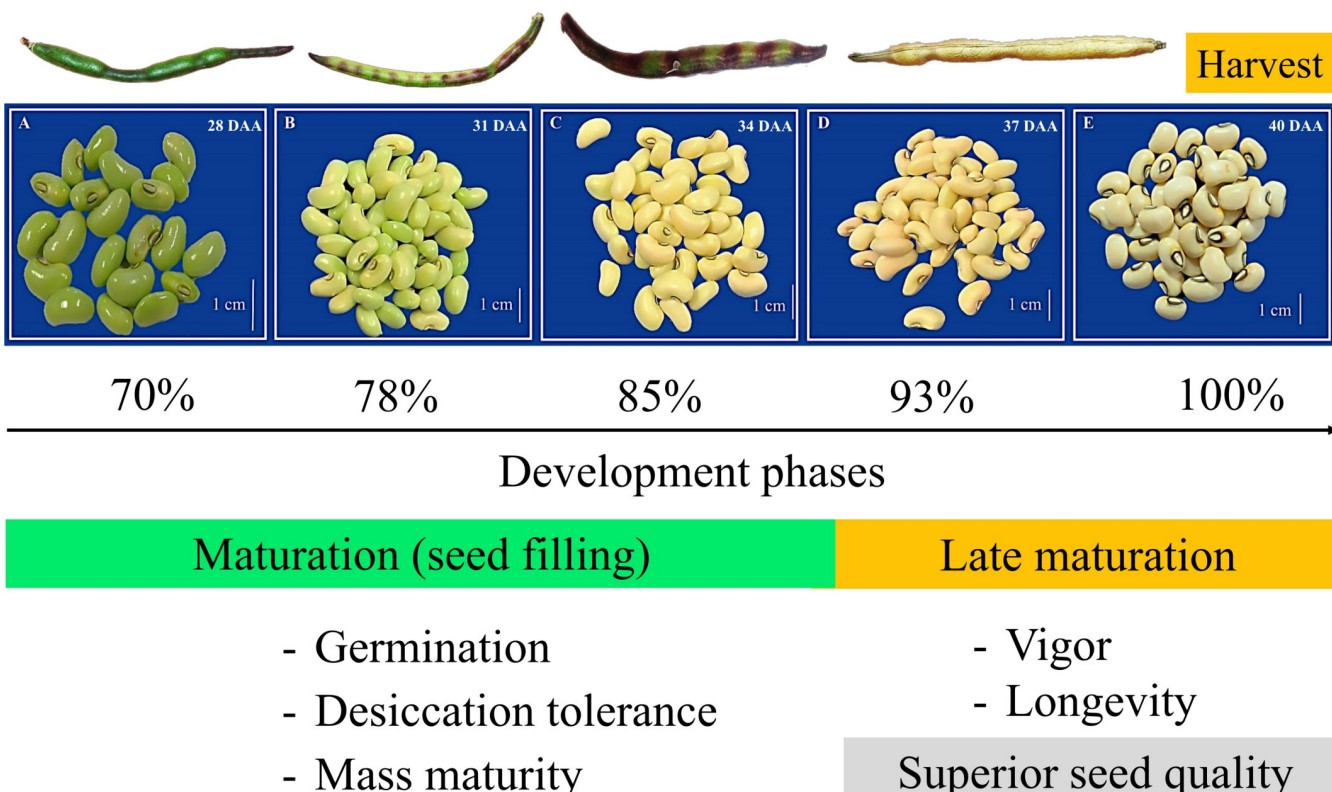

**Fig 4. Phases of cowpea seed development.** The relation between the developing stages and the acquisition of seed physiological quality during maturation and late maturation.

from 34 DAA (beginning of the late maturation stage) the seeds displayed a remarkable improvement in their ability to successfully install themselves in the agricultural environment. Noteworthy signs that determine the importance of seed vigor are seen in seedling performance [13], rapid germination [15] and superior potential to germinate after harvest and throughout the storage period (Fig 1B). Thinking about food production programs, harvesting cowpea seeds with maximum vigor can be strategic to expand the possibilities of success in the implantation of a future crop. This idea comes from the close relationship that seed vigor establishes with an adequate population of plants and increased grain yield [5, 9], even under climatic stresses [28]. In cowpea growing regions, having seeds with maximum physiological quality can mean a step forward in optimizing the supply of grains as a source of basic food for the population.

Considering logistics and scheduling for the effective use of cowpea seeds by farmers (seeds can be used months after the harvest), seed longevity is an essential part of food production programs. This attribute of physiological quality acquired at the end of development in late seed maturation (Fig 2B) is governed by mechanisms that minimize the deterioration process and extend shelf life in the dry state [12]. The formation of vitreous cytoplasm during seed drying [23] and the accumulation of compounds with antioxidant action in the maturation phase [26, 29] act as essential mechanisms of protection and repair. Both mechanisms delay deterioration reactions favoring maintenance of seed viability during storage [30] and preserve seed vigor [13]. By determining seed longevity by p50, information was obtained about their viability under stress conditions (35° C and 75% RH) that can easily occur in tropical regions. This

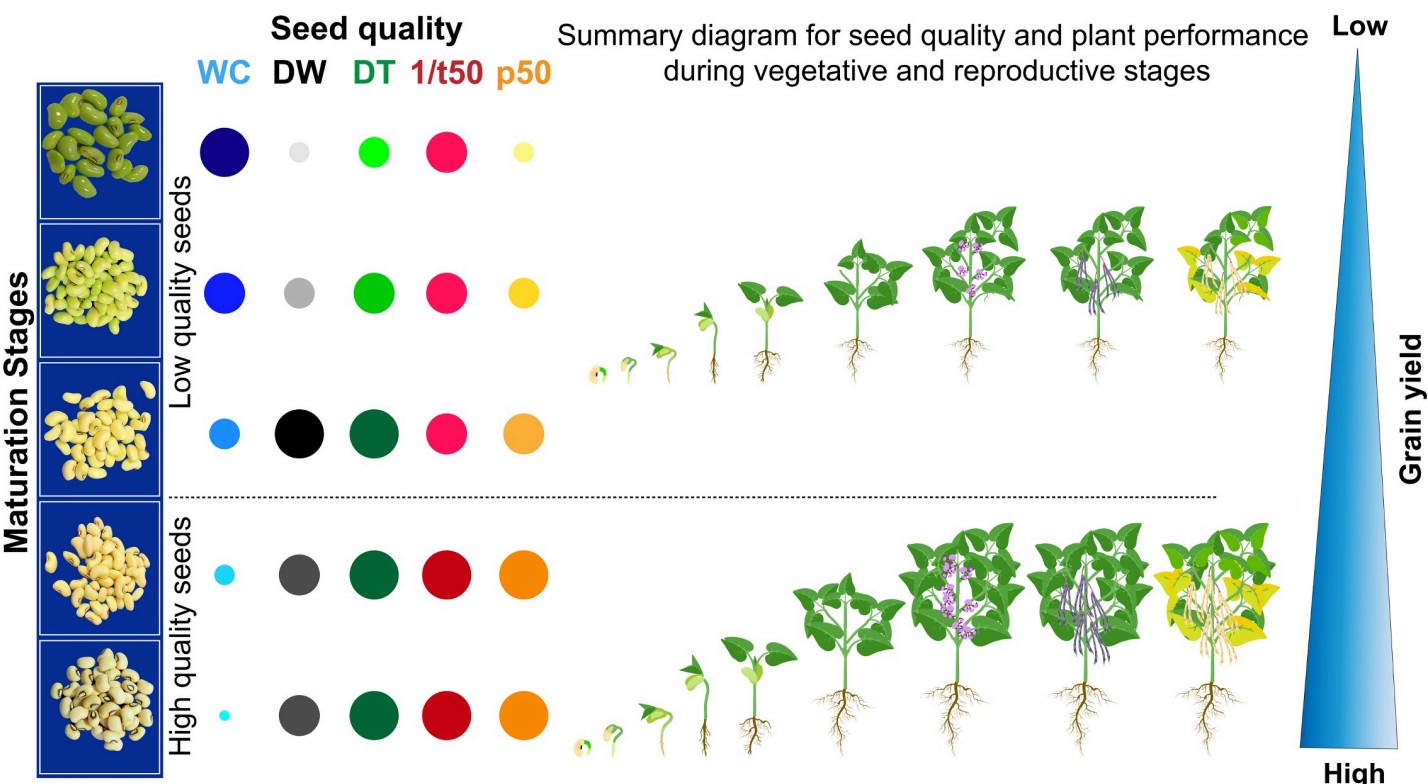

**Fig 5. Summary diagram for cowpea (*Vigna unguiculata* L., Walp) grain yield performance as a function of physiological seed quality at different maturation stages.**

reinforces the idea that the results presented here have a practical utility in post-harvest, especially in the agricultural context of nations lacking resources for proper seed storage. One may ask: what is the connection of this information to the reality of the cowpea production fields?

From the previous perspective, seed quality can play an essential role in food production and thus the livelihoods of communities around the world. The quality of cultivated species is the primary cause associated with mechanisms of plant life expression at the highest degree of sophistication. In our summary diagram, we express a probable result of the formation of the cowpea crop from seeds with different maturation stages and physiological quality levels (Fig 5). Supported by previous studies [5, 9, 28] we showed that grain yield starts from the high-performance of seeds capable of properly forming a more productive plant community. It is a fact that under stressful climatic conditions for plant cultivation, only the fittest living things (seeds) survive and multiply in the agricultural environment [31]. In this universal logic, only superior quality seeds have sufficient capacity to perpetuate the species in time, which in agricultural activity results in food. We present temporal fractions of cowpea seed development consistent with the behavior of documented orthodox species that govern the global food base. In regions inhabited by societies at constant nutritional risk, the use of seeds with superior quality can increase the supply of grains with high energy value. The main perspective of the proposed concept lies in its comprehensive applicability in agriculture and society.

## Conclusions

High quality seeds are a global food security strategy. In the context of cowpea production, harvesting seeds at the late maturation phase means accessing seeds with high physiological

quality and the highest potential for crop formation and grain yield. The practical effect of this process plays a key role in agriculture by contributing to food production. Our proposed summary diagram is an essential part of assertive planning to produce food in areas of frequent climatic stress. Therefore, seed quality is one of the indispensable factors for farmers to successfully grow agricultural species in tropical environments and essentially achieve food security and social sustainability.

## Supporting information

**S1 Fig. Daily rainfall, maximum and minimum temperatures in the experimental farm of Botucatu, State of São Paulo-Brazil, during cowpea seed production (2015 crop season).**
(DOCX)

**S2 Fig. Flower tagged and beginning of fruit development of cowpea (*Vigna unguiculata* L., Walp).** A: Open flower; B: Flower after 2 days of pollination; C: Fruit in formation 9 days after anthesis.
(DOCX)

**S1 Table. Information of the data evaluated in cowpea seeds (*Vigna unguiculata* L., Walp) during early maturation stages (17 to 25 DAA).**
(DOCX)

## Acknowledgments

We are thankful to Valeria Cristina Ratameiro Giandoni for her support during seed physiological analysis and to Mr. Roger Hutchings for the English review of the manuscript.

## Author Contributions

**Conceptualization:** Gustavo Roberto Fonseca de Oliveira, Thiago Barbosa Batista, João William Bossolani.

**Data curation:** Leticia de Aguila Moreno.

**Formal analysis:** Gustavo Roberto Fonseca de Oliveira.

**Funding acquisition:** Edvaldo Aparecido Amaral da Silva.

**Investigation:** Leticia de Aguila Moreno.

**Methodology:** Leticia de Aguila Moreno, Gustavo Roberto Fonseca de Oliveira, Thiago Barbosa Batista, João William Bossolani, Karina Renostro Ducatti.

**Project administration:** Leticia de Aguila Moreno, Edvaldo Aparecido Amaral da Silva.

**Resources:** Edvaldo Aparecido Amaral da Silva.

**Supervision:** Edvaldo Aparecido Amaral da Silva.

**Validation:** Gustavo Roberto Fonseca de Oliveira, Edvaldo Aparecido Amaral da Silva.

**Visualization:** Gustavo Roberto Fonseca de Oliveira, Thiago Barbosa Batista, João William Bossolani, Cristiane Carvalho Guimarães, Edvaldo Aparecido Amaral da Silva.

**Writing – original draft:** Gustavo Roberto Fonseca de Oliveira.

**Writing – review & editing:** Edvaldo Aparecido Amaral da Silva.

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
