## [Decision Letter · Decision Letter 0]

19 Jul 2022

PONE-D-22-16964Quality of cowpea seeds: A food security strategy in the tropical environmentPLOS ONE

Dear Dr. Amaral da Silva,

Thank you for submitting your manuscript to PLOS ONE. After careful consideration, we feel that it has merit but does not fully meet PLOS ONE’s publication criteria as it currently stands. Therefore, we invite you to submit a revised version of the manuscript that addresses the points raised during the review process.

We look forward to receiving your revised manuscript.

Kind regards,

Alessio Scarafoni

Academic Editor

PLOS ONE

Journal Requirements:

Additional Editor Comments:

Both reviewers raised numerous criticisms that must be resolved to reach the level required for the publication of the manuscript. I basically agree with them. The comments of the reviewer are very constructive and give some advice. I recommend the Authors to consider all of them with attention to improve the manuscript.

Reviewers' comments:

Reviewer's Responses to Questions

**Comments to the Author**

1. Is the manuscript technically sound, and do the data support the conclusions?

Reviewer #1: Partly

Reviewer #2: Yes

2. Has the statistical analysis been performed appropriately and rigorously? 

Reviewer #1: Yes

Reviewer #2: Yes

3. Have the authors made all data underlying the findings in their manuscript fully available?

Reviewer #1: Yes

Reviewer #2: Yes

4. Is the manuscript presented in an intelligible fashion and written in standard English?

Reviewer #1: Yes

Reviewer #2: Yes

5. Review Comments to the Author

Reviewer #1: This paper describes a useful study to assess the development of seed quality features in cowpea seeds during their maturation. It will add to a growing body of work showing that the last stage of seed development after maximum dry weight accumulation is important for the acquisition of maximum vigor and longevity. The authors note that this is particularly critical in warm, humid climates, in which seed longevity is shortened during storage under ambient conditions, potentially affecting stand establishment and yield in the following season.

In their abstract, the authors state that they “built an original theoretical model that links the development stages of the seeds with their potential of providing grain yield.” When I originally read this, I expected some type of novel quantitative or predictive model based on and tested by their results. However, this model is a diagram illustrating the development of seed quality on the one hand and corresponding illustrations of expected plant growth from those seeds, which was not actually tested in this paper. Based on multiple papers cited by the authors, this is very likely to be the case, and is neither theoretical nor original. These issues are also discussed, for example, in classic texts such as D.B. Egli. 2017. Seed Biology and Yield of Grain Crops. 2nd Edition. CABI. Thus, I would remove those terms referring to such a model and simply refer to Figure 5 as a summary diagram relating their work specifically on cowpea to other work (e.g., refs 7, 9, 13, 15). However, the diagram needs to be modified a bit to be accurate. As the authors showed, the time to 50% germination is much longer in less mature seeds than in more mature seeds. This should be illustrated in Fig. 5 by shifting the upper illustration to the right, indicating the slower emergence and start to vegetative development compared to the more mature seeds. The illustrations of the growth stages could be crowded together more or perhaps a stage removed such that maturity occurs at the same time but the plants are smaller. The later emergence thus would result in a shortened growth period and reduced yield. This would be consistent with the illustrations in ref 9, which attribute the poorer growth of late emerging seedlings to the shading by plants that emerged earlier. This turns out to be the major effect of seed vigor on crop production, i.e., variability in emergence times and reduced plant populations, rather than reduced potential growth rate per se of the resulting plants. Earlier harvests would naturally increase variation in developmental maturity within the seed lot, in contrast to the situation here with tagged flowers harvested at specific times, so that the consequences illustrated in refs. 7, 9 of mixed quality lots would occur.

In addition to these primary recommendations, there are also a number of minor corrections and edits that should be done prior to publication as listed below.

1. Line 159: Figure 5 does not really resemble a Heatmap to me. The size and color of the circles do relate to the changes in the parameter values, although this is rather confusing with respect to t50. While it is correct that t50 decreases during maturity, this actually represents higher quality, rather than less, as the smaller circles and lighter color would seem to indicate. Instead, I would recommend using the germination rate, or 1/t50, which increases with seed quality and is actually normally distributed in a seed population (e.g., Hay et al. (2014) Seed Science Research 24, 165-186). This might also impact the PCA analysis, as it currently shows high t50 associated with immature seeds, which is correct, but it does not show low t50 being associated with later development, or higher quality. Using 1/t50 in these analyses would correct these issues.

2. Line 186: Stated here that germination capacity reached a maximum at 31 DAA, whereas it appears to be at least 34 or 37 DAA before the curve peaks.

3. Line 187-189: This statement contradicts the previous one, which said that germination capacity peaked at 31 DAA, yet here it says that seed quality continued to increase after maximum DW, which was at 34 DAA.

4. Line 192: This statement does not agree with the data in Fig. 1B, where germination after desiccation at 28 DAA was 0%, not 23% as stated in this text (I did not have access to supplemental data).

5. Line 198-199: It should be noted again here that the normal seedlings were in the early count of the germination test, not the final count. Delayed germination will reduce the percentage of normal seedlings in an early count, but may not affect the total number of normal seedlings after a longer imbibition time.

6. Line 201: Do you mean Fig. 1A and 1B here?

7. Line 202: Do you mean Fig. 2A here? There is no Fig. 2D.

8. Line 202-3: According to Methods, the seeds were previously dried before the aging test, and dried seeds did not germinate after drying at 28 DAA (Fig. 1B). How then is the initial germination percentage of 28 DAA seeds about 40% in Fig. 2A?

9. Lines 203-205: The storage times mentioned here are when the last seed died. In general, longevities are referred to with respect to a specific percentage, e.g., 50% or the p50. However, the p50 values cited in the following sentences (up to 178 days) do not correspond with the p50 values that would be derived from Fig. 2A. For example, the p50 in Fig 2B for 31 DAA is about 60, whereas in Fig 2A, the curve for 31 DAA would cross 50% at about 40 days. Something is not correct about how p50 values were estimated in 2B from the data in 2A. I hope that the germination capacity values were all based on the total seed population tested, and not only on the number of viable seeds. This may be the case, as Fig. 1B shows germination capacity of 31 DAA seeds being about 20% after drying, while the curve for 31 DAA seeds starts near 100% in Fig. 2B. In longevity studies, the percentage has to be the fraction of total seeds. If the developing seed population had not reached 50% germination capacity, then it is not possible to estimate p50 for direct comparison with more mature seeds. While it may not affect the overall results of the analysis, the PCA should be run again using corrected values for p50s.

10. Line 273: The temperature of aging is given as 35C in the Methods.

11. Line 278: Awkward phrasing. The last two sentences of this paragraph could likely be deleted.

12. Line 300: See comments above about “proposed theoretical model”, which is not original nor theoretical, and there is substantial prior data to support it. You can still make your concluding case and summarize your results in Fig. 5 without claiming that it is an original theoretical model.

Reviewer #2: To author:

The manuscript entitled "Quality of cowpea seeds: A food security strategy in the tropical environment" is a study that contributes to the area of production of quality of seeds. Although the topic is well studied in the literature some results presented in the study are relevant. Furthermore, the results presented support the study hypothesis. However, some clarifications are required that will contribute to the presentation manuscript, as follow in the report.

Major compulsory Revisions:

Page 9, line 77 to 82 – In the sentence “This data allowed us to define the moment to Harvest cowpea seeds with the highest quality. It also allowed the construction of a theoretical model with application to agricultural species propagated by orthodox seeds. In this model, we connected the phases of acquisition of physiological quality with the establishment of plants in the field and grain yield. The knowledge acquired in this work, added to that existing in the literature, brings the idea that enhancing the quality of cowpea seeds increases its relevance in food security strategies”, it must be included in the topic of discussion or excluded.

Page 11, line 120 to 163 – Authors should consider including subtopics to present “Seed quality assessment” (water content of fresh; germination capacity; determination of longevity; maturation and late maturation stages)

Page 18, line 295 to 303 – Authors must present a conclusion considering the evidence based on the results found in the study.

Some minor criticisms are included below:

Page 14, line 202 – Revise the indication “Fig 2D”.

Page 22, Table 1 – Adjust the indication “*”. Put in the final of title of the table.

“Table 1. Statistical Analysis. Information of the data evaluated for cowpea seeds (Vigna unguiculata L., Walp) during maturation stages (28 to 40 DAA).*”

Page 22, Table 1 – Remove the indication “1”.

6. PLOS authors have the option to publish the peer review history of their article (what does this mean?). If published, this will include your full peer review and any attached files.

Reviewer #1: No

Reviewer #2: No

---

## [Author Response · Author response to Decision Letter 0]

26 Sep 2022

Manuscript Number: PONE-D-22-16964

Quality of cowpea seeds: A food security strategy in the tropical environment

Alessio Scarafoni PhD,

Academic Editor of PLOS ONE

 Dear Dr. Scarafoni,

We thank you and the reviewer for the corrections and suggestions made in the manuscript. Below, we provide our answers point-by-point to the comments and suggestions made.

With warm regards,

Amaral da Silva (corresponding author on behalf of co-authors)

Reviewer #1: 

Answer: The term "theoretical model" has been duly corrected as suggested. Regarding the diagram, the germination rate was added in place of t50. We would like to point out that the images illustrate the cycle of a plant coming from seed with low quality (immature seed) and high physiological quality (mature seed) and not a plant stand. So, if we understand the suggestion, the diagram is adequate to illustrates the proposal of the work. If there are more to correct, we are at your disposal to improve the manuscript.

Detailed comments

1 and 2. Answer: As the reviewer suggested, Figures 1, 3, and 5 have been modified. Thanks.

2. Answer: The correction has been made (lines 206 and 207). Thanks.

3. Answer: The correction has been made (lines 201). Thanks.

4. Answer: The correction has been made (line 200). Thanks.

5. Answer: The evaluation of normal seedlings at five days made it possible to identify the capacity for complete formation of embryonic tissues early. Only the most vigorous seeds have this capacity. Therefore, we consider that for vigor evaluation, five days was enough to detect what we were looking for. We thank you for the suggestions and we are available for further discussion. Thanks.

6. Answer: Yes. The corrections have been made (line 216). Thank you very much.

7. Answer: Yes. The corrections have been made (line 217). Thanks.

8. Answer: Thank you very much for this observation. The data has been duly reviewed (Figs 1 and 2) and corrections made (lines 208 to 215).

9. Answer: All the suggested corrections have been made. Thank you very much (lines 216 to 227).

10. Answer: The correction has been made (line 290). Thanks.

11. Answer: The correction has been made (line 294). Thanks.

12. Answer: The correction has been made (Fig 5). Thanks.

Reviewer #2: 

1. Answer: Based on the data obtained, the constructed diagram allowed us to raise the idea of the use of high quality seeds as a food security strategy. Therefore, we reinforce that the discussion of this topic is based on the last paragraph of the manuscript (lines 296 to 310).

2. Answer: The correction has been made. Thanks.

3. Answer: We started the conclusion based on the results of the study. Afterwards, we end with a general idea built on these scientific results. We emphasize that the conclusion is not only the concrete expression of the results, but a general interpretation of them with a broader scope. We propose that the seed is the link to food security and has the potential to be used by multi areas in science. Therefore, we consider that our conclusion is correct. However, we are happy to improve it if more corrections are necessary.

4. Answer: The correction has been made (line 218). Thanks.

5. Answer: The correction has been made (line 218). Thanks.

6. Answer: The correction has been made (line 218). Thanks.

7. Answer: The correction has been made (line 218). Thanks.

8. Answer: The correction has been made (line 218). Thanks.

---

## [Editor Report · Decision Letter 1]

29 Sep 2022

Quality of cowpea seeds: A food security strategy in the tropical environment

PONE-D-22-16964R1

Dear Dr. Amaral da Silva,

We’re pleased to inform you that your manuscript has been judged scientifically suitable for publication and will be formally accepted for publication once it meets all outstanding technical requirements.

Kind regards,

Alessio Scarafoni

Academic Editor

PLOS ONE

Additional Editor Comments (optional):

The authors substantially revised the manuscript in accordance with the comments and recommendations of the reviewer. The answers to the comments and questions that both Reviewers raised have been clearly addressed and reported in the text and in the figures. The Authors made most of the changes that have been suggested. The rebuttals to a couple of points made by the authors are embraceable and are justified following responses and changes to other comments raised. Contradictions that occurred in the original version have been resolved. The text is now clearer and fluid and the significance of the findings is now expressed and commented. The readability for either expert in the field and for the general audience greatly improved.

I recommend the publication of the manuscript in its present form.
---

## [Editor Report · Acceptance letter]

5 Oct 2022

PONE-D-22-16964R1 

Quality of cowpea seeds: A food security strategy in the tropical environment 

Dear Dr. Amaral da Silva:

I'm pleased to inform you that your manuscript has been deemed suitable for publication in PLOS ONE. Congratulations! Your manuscript is now with our production department. 

Kind regards, 

on behalf of

Dr Alessio Scarafoni 

Academic Editor

PLOS ONE